# Influence of Diminished Ovarian Reserve on Early Embryo Morphokinetics during In Vitro Fertilization: A Time-Lapse Study

**DOI:** 10.3390/jcm11237173

**Published:** 2022-12-02

**Authors:** Lisa Boucret, Léa Tramon, Jérémie Riou, Véronique Ferré-L’Hôtellier, Pierre-Emmanuel Bouet, Pascale May-Panloup

**Affiliations:** 1Reproductive Biology Unit, Angers University Hospital, 49000 Angers, France; 2Delegation for Clinical Research and Innovation, Angers University Hospital, 49000 Angers, France; 3Department of Reproductive Medicine, Angers University Hospital, 49000 Angers, France; 4MITOVASC, INSERM 1083, CNRS 6015, Angers University, 49000 Angers, France

**Keywords:** diminished ovarian reserve (DOR), ovarian aging, time-lapse imaging, morphokinetic assessment

## Abstract

There is great controversy as to whether women with Diminished Ovarian Reserve (DOR) exhibit only a quantitative decrease in ovarian reserve or also impaired oocyte and embryo quality. In this retrospective study, we aimed to evaluate the impact of DOR on embryo morphokinetic parameters with a time-lapse system. 1314 embryos were obtained from 256 couples undergoing IVF or ICSI cycles, with 242 embryos in the DOR group as classified by the Bologna and POSEIDON criteria and 1072 embryos derived from the Normal Ovarian Reserve (NOR) group. For each morphokinetic parameter (t2, t3, t4, t5, t8, tB, ECC2, cc2a, ECC3, s2, s3), a generalized linear mixed model was created to control for female age, BMI, smoking status, method of insemination and correlation between oocytes from a same cohort. No significant association was found between DOR and any of the morphokinetic parameters studied. In a secondary analysis, we evaluated the influence of maternal aging, comparing morphokinetic characteristics between two age groups (<37 and ≥37 years). In the univariate analysis, we found that embryos from older women displayed a slower embryo development (in particular for t3, t4, t5, tB, and ECC2), although without statistical significance in the multivariate analysis. In conclusion, our study did not reveal any substantial impact of ovarian aging on early morphokinetic parameters and suggested potential biases that may be a source of controversy in the literature.

## 1. Introduction

The fecundity of women declines dramatically with advancing age. However, cultural and social trends have led women in recent decades to postpone childbearing into their 30s and, in some cases, their 40s. The average age of women at first birth in the United States has increased from 21.4 in 1970 [1] to 27.0 in 2019 [2]. In today’s world, environmental pollutants and modern lifestyles represent additional factors that may contribute to accelerated ovarian aging and infertility.

Diminished ovarian reserve (DOR) is a concept tightly intertwined with ovarian aging and refers to women of reproductive age having regular menses whose response to ovarian stimulation or fecundity is reduced [3]. The prevalence of DOR among cycles from the Society for Assisted Reproductive Technology (SART) registry is estimated to reach 26% [4]. Despite advances in stimulation protocols and In Vitro Fertilization (IVF) techniques, DOR represents a major challenge in reproductive medicine. Indeed, DOR patients are at risk of a poor response to ovarian stimulation, higher cycle cancellation rates, and a decrease in the number of oocytes obtained, which dramatically reduces their chances of conceiving. The IVF success rate in women with poor ovarian response is estimated at 6% live births per cycle [5].

Unfortunately, the exact pathophysiology underlying DOR remains poorly understood. Despite the extensive literature reviewing the clinical management of patients with DOR, there is still no consensus regarding whether their poor prognosis in IVF is attributed only to lower oocytes retrieval or whether it also reflects impaired oocyte and embryo quality. This comprehension is limited by heterogeneous definitions of DOR, methodologic discrepancies in selecting proxy indicators reflecting oocyte quality, which cannot be directly assessed, as well as the difficulty in differentiating physiologic from premature declines in ovarian function.

In recent years, time-lapse microscopy (TLM) has been introduced, allowing continuous monitoring of embryo development without the need to disturb culture conditions by removing embryos from the incubator. This non-invasive approach records a large amount of morphokinetic data, which could not necessarily be collected with conventional snapshot observations, and thus helps embryologists to assess embryos for identifying those with the highest implantation potential. It is hypothesized that accurate timings of particular events, such as early cleavage [6], cell cycle intervals [7,8,9], the synchronicity of specific cell divisions [10], and the start of blastulation [11], are associated with pregnancy outcomes. To the best of our knowledge, only scarce studies have explored morphokinetic parameters of embryos derived from patients with DOR [12,13,14]. There is also little information on the effect of maternal age on morphokinetic timings [15,16,17]. TLM could enable optimal embryo selection in DOR women, which could improve their assisted reproductive technology (ART) outcomes. Therefore, the aim of the current study was to compare absolute and relative morphokinetic parameters of embryos obtained from women with and without DOR in order to evaluate the impact of DOR on morphokinetic patterns of embryo development.

## 2. Materials and Methods

### 2.1. Study Design and Population

This was a single-center retrospective cohort study that included IVF or ICSI cycles performed in the reproductive medicine center at Angers hospital (France) from January 2020 to May 2021. Ovarian reserve was evaluated according to age, baseline follicle-stimulating hormone (FSH) level, antral follicle count (AFC), and anti-Mullerian hormone (AMH) level. Based on Bologna [18] and POSEIDON [19] classification, at least two of the following criteria had to be present to define DOR:Advanced maternal age (≥40 years);≤3 oocytes retrieved in the current or a previous cycle;An abnormal ovarian reserve test (i.e., AMH < 1.2 ng/mL and/or AFC < 7 follicles, and/or FSH ≥ 10).

To be categorized in the NOR group (control group), women did not have to meet more than one of the aforementioned criteria.

All women were without known risk factors for genetic, surgical, or iatrogenic ovarian insufficiency. Other exclusion criteria were as follows: (1) Polycystic ovary syndrome (PCOS); (2) Endometriosis; (3) Egg donation; (4) Age below 18 and over 43 years (age limits for access to IVF treatment in France); (5) Opposition of the patient to the processing of his personal data; (6) No embryos available for transfer.

### 2.2. Controlled Ovarian Stimulation

IVF stimulation protocols were selected by primary physicians based on AFC, AMH, age, body mass index (BMI), and ovarian response in previous IVF cycle(s). The majority of cycles (91%) used GnRH antagonists after estrogen priming. In the second approach, pituitary suppression was performed with a gonadotrophin-releasing hormone agonist. Ovarian stimulation was achieved using recombinant FSH and/or human menopausal gonadotrophin, with doses ranging from 125 to 450 IU per day. Ovarian response was monitored via transvaginal ultrasonography and serum estradiol (E2) measurements. Triggering was performed with 6500 IU of recombinant human chorionic gonadotropin (rhCG) and/or triptorelin 0.2 mg when at least three follicles achieved a 17 mm diameter. Oocytes were retrieved 36 h later.

### 2.3. Fertilization

Retrieved oocytes were washed with Flushing medium (CooperSurgical^®^, Ballerup, Denmark) and placed for 1–3 h in FertiCult^®^ IVF medium (FertiPro, Beernem, Belgium) at 6% CO_2_, 5% O_2_, and 37 °C. Semen was collected and processed following the World Health Organization (WHO) guidelines [20]. Samples were classified according to the prewash total motile sperm count (TMSC) (cut-off value of 5 million). Semen preparation was performed using a standard gradient separation (PureSperm^®^, Nidacon, Mölndal, Sweden) at 300× *g* for 20 min, followed by washing at 600× *g* for 10 min. Oocytes were inseminated using either conventional insemination procedures (IVF) or micromanipulation techniques (ICSI) according to the center’s guidelines. Conventional insemination was performed with 100,000 motile sperm in 1 mL of FertiCult^®^ IVF medium containing up to five cumulus-oocyte complexes (COCs). COCs were denuded the following day (19–20 h after insemination), and fertilization was determined. For ICSI cycles, hyaluronidase (80 IU/mL) was used to denude oocytes from cumulus cells at least one hour after oocyte retrieval. Microinjection was performed on mature oocytes at 400× magnifications using an Integra Ti™ Micromanipulator (Cooper Surgical Company, Trumbull, CT, USA).

### 2.4. Embryo Culture

Embryos were cultured individually in preequilibrated EmbryoSlides (Vitrolife) using a single-step culture medium (LifeGlobal, Ballerup, Denmark) supplemented with 10% of Human Serum Albumin (HSA) (Cooper-Surgical, Ballerup, Denmark) in an EmbryoScope+ time-lapse incubator (Vitrolife, Västra Frölunda, Sweden). The tri-gas TLS incubator was set to 5% O_2_, 6.0% CO_2_, and 37 °C.

### 2.5. Time-Lapse Imaging System and Analysis

Embryo images were recorded every 10 min in 11 different focal planes. Morphokinetic parameters were annotated by two investigators on the EmbryoViewer^®^ Software 7 (Vitrolife, Västra Frölunda, Sweden). Morphokinetic events and calculated variables were assessed according to Ciray’s guidelines [21]. Briefly, morphokinetic timings included: time to syngamy (tPNf), time to two (t2), three (t3), four (t4), five (t5), and eight (t8) cells, as well as time to blastocyst (tB). Timings of cell divisions were measured in hours from time zero (t0), which was assigned as pronuclei fading to eliminate ambiguity regarding the use of insemination or injection timing. Calculated variables related to the dynamics of the early preimplantation period included: the duration of the second cell cycle (ECC2 = t4 − t2), in particular for blastomere ‘a’ (cc2a = t3 − t2), duration of the third cell cycle (ECC3 = t8 − t4), the synchronicity of the two blastomere divisions within the second cell cycle (s2 = t4 − t3), and synchronicity of the four blastomere divisions within the third cell cycle (s3 = t8 − t5).

### 2.6. Embryo Transfer

Embryos were selected for transfer according to a combination of factors: normal fertilization (embryos with 2 pronuclei), KIDScore™ result, standard morphological grading, and morphokinetic exclusion criteria (direct or reverse cleavage). One or two embryos were transferred under transabdominal ultrasound guidance at the cleaved or blastocyst stage depending on patient demographics (age, gynecological history, cycle rank) and embryo development (number and quality of available embryos). Luteal phase support was started on the day of oocyte retrieval with oral dydrogesterone (30 mg/day) and continued until 12 weeks of gestation if a positive pregnancy test (HCG >100 IU/L) was obtained 14 days after oocyte retrieval.

### 2.7. Data Collected and Outcome Measures

Information was collected from the medical charts on baseline characteristics (age, BMI, smoking), ovarian reserve (AMH, CFA, FSH, E2, and luteinizing hormone (LH) at day 3), ovarian stimulation parameters (cycle rank, agonist or antagonist protocol, gonadotropin type, total dose of FSH administered), as well as IVF cycle parameters (number of oocytes and embryos collected, TMSC, embryo stage at transfer, number of embryos transferred), and morphokinetic parameters (tPNf, t2, t3, t4, t5, t8, tB, ECC2, cc2a, ECC3, s2, s3).

### 2.8. Statistical Analysis

Descriptive statistics were calculated according to the DOR status. Categorical variables were summarized with the use of counts and percentages and continuous variables with mean values and standard deviations (SD). Continuous variables were compared using the Student’s *t*-test and qualitative variables were compared with the Chi-squared test. A secondary analysis was performed in which the ovarian reserve was evaluated on the basis of the AMH value only, as this latter seems to be one of the best tests of ovarian reserve currently available [22,23,24]. Finally, we also classified the women into age groups. A cut-off of 37 years old was chosen because of a sudden acceleration of follicle loss at this age [25,26].

The multivariate analysis consisted of generalized linear mixed models with time to each stage or duration of events as the outcomes (t2, t3, t4, t5, t8, tB, ECC2, cc2a, ECC3, s2, s3). To explicitly control for differences in the patient populations, potential confounding variables (age, BMI, smoking, and method of insemination) were considered as other explanatory variables in the models. A random effect was included to account for the clustered nature of the data (correlation between embryos derived from the same cycle). The assumption of normal distribution of residuals was checked, as assessed by Q-Q plots. When necessary, the most appropriate box/cox transformation was identified and applied to the outcome variable. All analyses were performed using the statistical software package R version 4.1.1 [27]. A *p*-value of <0.05 was considered statistically significant.

### 2.9. Ethical Approval

The study was conducted in accordance with the Declaration of Helsinki and approved by the hospital’s Ethics Committee (University Hospital of Angers, France, reference number 2021-214). The anonymized database was registered under CNIL approval number ar22-0007v0. All patients gave informed consent for the use of their anonymized clinical data in the observational study.

## 3. Results

During the period, 256 couples undergoing IVF or ICSI cycles were included in this retrospective study, resulting in 1314 embryos. Among them, 242 embryos were derived from the DOR group (76 women), and 1072 embryos were obtained from the NOR group (180 women).

### 3.1. General Characteristics of Patients and IVF Cycles Parameters

The baseline characteristics of the two groups are shown in Table 1. In the DOR group, the values of AMH and AFC were significantly lower, and the FSH level was significantly higher than in the NOR group (*p* < 0.01), as expected because these values were the criteria used for classifying the groups. Likewise, DOR patients were significantly older than NOR patients (*p* < 0.01). There was no significant difference in terms of BMI (*p* = 0.1) or tobacco use (*p* > 0.9). With regard to the ovarian stimulation treatments (Table 1), the total dose of FSH administered was significantly higher in the DOR group compared with the NOR group (*p* < 0.001). Patients with DOR were more likely to receive FSH combined with LH (*p* < 0.01), whereas there was no significant difference in the pituitary suppression protocol between the two groups (*p* = 0.1).

The IVF cycle parameters are described in Table 2. As one would expect, male infertility was more common in the NOR group, with a significantly lower TMSC (*p* = 0.03) and more ICSI performed (*p* = 0.006). Fewer oocytes and embryos were obtained from DOR patients (*p* < 0.001) when compared to the control group. Double embryo transfers were significantly more frequent in the DOR group (*p* = 0.03), whereas there was no statistically significant difference between the two groups with regard to the day of transfer (*p* = 0.6).

### 3.2. Morphokinetic Analysis

#### 3.2.1. Comparative Analysis between the DOR and the NOR Groups

The timing of each cellular stage, as well as the duration and the synchronicity of the cell cycles, were compared between the NOR and the DOR groups (Table 3). Embryos from the two groups displayed overall similar developmental patterns (*p* > 0.05).

We also considered the ovarian reserve according to the AMH level only. AMH level did not significantly influence any of the morphokinetic parameters evaluated (*p* > 0.05).

#### 3.2.2. Comparative Analysis by Age Group

The absolute and relative morphokinetic events were next compared between different age groups (Table 4). Embryos from women ≥37 years old reached the 3-cell, 4-cell, 5-cell, and blastocyst stages significantly later than embryos from younger patients (*p* = 0.02, *p* = 0.006, *p* = 0.04 and *p* = 0.008 respectively). In addition, maternal age significantly affected ECC2 (*p* = 0.02).

A mixed-effects model was performed to control for BMI, smoking, and method of insemination and to account for the correlation between embryos from the same patient cohort. After adjustment, morphokinetic parameters were no longer significantly different between the two age groups (Table 5).

## 4. Discussion

This study presents an extensive analysis of the preimplantation embryo development in a cohort of embryos from DOR women. We failed to demonstrate the influence of DOR on any specific cellular timing, either by classifying patients according to the Bologna and Poseidon criteria or according to the AMH value alone. When we categorized the patients according to their age, using a cut-off of 37 years [25,26], we found that embryos from older women displayed significantly longer t3, t4, t5, tB, and ECC2 when compared to the younger group, but this difference was no longer significant after adjustment for confounding factors.

The available literature relative to the morphokinetic parameters in patients with diminished ovarian reserve is limited and heterogeneous. Akarsu et al. [12] reported no differences between DOR and NOR patients on the early embryo morphokinetic parameters but noticed that some of these (tPNf, t2, t3, t4) were significantly shorter in the younger age group of NOR patients. A Danish retrospective study described a trend for a slower t3 in women with poor ovarian response (AMH ≤ 1.1 ng/mL and number of oocytes retrieved <7) when compared to controls after adjustment for age [28]. Disparities in the experimental design of the different studies (differences in the definition of the DOR and the control groups and in the fertilization methods) might explain the apparently divergent conclusions.

For some authors, in the context of ovarian aging, it would be important to distinguish biological age from chronological age [29,30]. In the second part of this study, we considered only the chronological age in order to confirm our results under both hypotheses. Only a few studies have investigated the influence of maternal age on early embryo development and, in particular, on cleavage patterns. We noticed in the univariate analysis that embryos from younger patients developed overall faster than those from older patients, a phenomenon also described in a well-designed study published by Barrie et al. [31]. This difference was more pronounced for time to blastulation. Barrie et al. hypothesized a cumulative delay in embryo development that becomes apparent by the time the blastulation stage is reached. Likewise, Kirkegaard et al. [32] suggested that the blastocyst parameters are more prone to be affected by patient-related factors than cleavage stage parameters.

More specifically, Akhter et al. noticed that t4 and t5 were delayed in women over 40 years old as compared to young patients [15]. Recently, Faramarzi et al. [16] reported that only t5 occurred later among women aged 36–40 and >40 years when compared to younger women (<30 and 30–35 years), whereas other morphokinetic parameters were comparable between the groups. In the two aforementioned studies as well as in the present study but before adjustment, maternal age seems to play a crucial role on t5, which itself appears to be one of the most predictive parameters for subsequent implantation and is incorporated into some time-lapse algorithms [10]. However, once again, there exists some controversy about the impact of maternal age on embryo morphokinetics. For example, Warshaviak and al. observed no significant differences in the morphokinetic parameters during the first 3 days of pre-implantation development between embryos of advanced-age women (≥42 years) and those of the control group [17].

In another field, our results chime with a previous study published by Schachter-Safrai et al. [13], who explored the morphokinetic parameters of women with a decreased ovarian response to controlled stimulation (≤5 retrieved oocytes). They revealed that t3, t4, tB, and ECC2 are potential key morphokinetic milestones, as they occurred significantly later in the poor responder group. In our study, we found that these specific timings were significantly delayed in the oldest age group. In both cases, these results were obtained using univariate analysis, but when we controlled for other confounders in a multivariable regression analysis, age was no longer independently associated with the timing of these morphokinetic events. As mentioned by Kirkegaard et al. [32], the results of the univariate tests should be interpreted with caution because they do not take into account the clustering of embryos within a same cycle and carry a high risk of overestimating the correlations.

Direct cleavage, first described by Rubio et al. [7] as a two- to three-cell cleavage occurring in <5 h, is a phenomenon well known to affect implantation potential [7,8,9,33]. Our study did not reveal significant differences in the duration of cc2a (t3 − t2) between age groups. In the same way, the incidence of direct cleavage described by Zhan et al. was similar between the different maternal age groups [8]. We also found that DOR did not significantly influence cc2a. Similarly, a recent study failed to establish a statistical association between DOR and the occurrence of rapid cleavage [34]. Based on the hypothesis of Zhan et al. [8] that one possible mechanism underlying direct unequal cleavage is the formation of multipolar spindles through the introduction of either incomplete, defective, or supernumerary centrioles by defective sperm, it would be interesting in a future study to investigate the influence of the male partner age and sperm defects on direct cleavage.

It is well known that female aging has detrimental effects on ovarian response to stimulation, cancellation rates, and required doses of gonadotropins [35,36]. Women of advanced age are more prone to a decrease in the number of oocytes retrieved [35], a decrease in clinical pregnancy and live birth rates [35,37,38], an increase in miscarriages [39], and congenital anomalies [40]. Aneuploidy is an important factor underpinning these pejorative clinical outcomes since aneuploidies increase dramatically with maternal age [41,42]: from 52% in blastocysts of women ≤32 years old, to 70% in women 37–41 years old, up to 90% in women ≥42 years old [43]. Molecular and cellular evidence is accumulating regarding the effect of ovarian aging: mitochondrial dysfunction [44], telomere shortening, cohesin dysfunction, spindle instability, alteration of gene expression patterns, and epigenetic modifications [39]. However, despite all of this, our study did not show that the detrimental impact of age on oocyte quality was characterized by a change in the morphokinetic parameters of the preimplantation embryo.

This study should be considered in light of its strengths and limitations. Strengths include its relatively large volume of embryos obtained in a single academic center with a consistent approach to clinical and laboratory management of cycles. By taking two alternate approaches to define DOR, the robustness of our findings was strengthened. Our statistical modeling adequately addressed issues regarding confounding age and the clustered nature of the data. Nonetheless, the limitations of our study should be pointed out. Because of its retrospective nature, we are not able to assume that all confounding factors were accounted for. Indeed, Kirkegaard et al. suggested that a high part of the fluctuations in morphokinetics is patient-dependent and that the observed variations are most likely caused by a combination of several factors rather than a single factor responsible for a systematic influence [32]. Strict criteria for patient inclusion were used to mitigate this limitation.

These preliminary results did not reveal any substantial impact of ovarian aging on early morphokinetic parameters. It could be interesting to extend this analysis with larger prospective studies. It would also be relevant to investigate the potential influence of aging and DOR on the KIDScore™ result, whose algorithm integrates several early morphokinetic events. To the best of our knowledge, only one study assessed the association between the serum AMH level, as a marker of ovarian reserve, and the KIDScore™ generated by TLS as an indicator of embryo quality [45]. This study showed no significant difference in mean serum AMH levels between the different KIDScore™ categories.

## 5. Conclusions

In conclusion, our data provide insight into a strictly defined DOR cohort and emphasize that DOR does not sharply negatively affect the timing of the preimplantation embryo development. Moreover, although embryos from patients with advanced maternal age seemed to display a slower embryo development, our results did not reach statistical significance when potential confounders were accounted for. Our study suggests that the heterogeneity of existing data in the literature concerning the impact of ovarian aging on embryo morphokinetics might be the result of insufficient consideration of confounding factors in the design of these studies.

## Figures and Tables

**Table 1 jcm-11-07173-t001:** Baseline characteristics and IVF stimulation parameters of Diminished Ovarian Reserve (DOR) versus Normal Ovarian Reserve (NOR) patients. Values are shown as mean ± SD or *n* (%).

	DOR (*n* = 76)	NOR (*n* = 180)	*p*-Value
Female age (year)	36.6 ± 4.1	33.9 ± 4.6	<0.001
BMI (kg/m^2^)	21.8 ± 2.8	22.5 ± 3.2	0.1
Smoking status			>0.9
Never	55 (73.3%)	131 (73.2%)	
Current	12 (16.0%)	28 (15.6%)	
Former	8 (10.7%)	20 (11.2%)	
AMH (ng/mL)	0.9 ± 0.4	2.9 ± 1.5	<0.001
AFC	11.0 ± 4.9	19.0 ± 6.8	<0.001
Basal FSH (IU/L)	10.8 ± 4.6	7.1 ± 1.6	<0.001
Basal LH (IU/L)	6.7 ± 6.0	5.8 ± 3.5	0.2
Basal E2 (pg/mL)	43.4 ± 18.1	36.0 ± 12.7	0.0003
Paternal age (year)	38.2 ± 5.6	35.9 ± 5.3	0.002
Paternal smoking status			0.4
Never	37 (52.9%)	96 (61.9%)	
Current	14 (20.0%)	40 (25.8%)	
Former	19 (27.1%)	19 (12.3%)	
Stimulation Protocol			0.4
GnRH Antagonist	67 (88.2%)	165 (91.7%)	
GnRH Agonist	9 (11.8%)	15 (8.3%)	
Gonadotropin type			<0.001
FSH	35 (46.1%)	132 (73.3%)	
FSH + LH	41 (53.9%)	48 (26.7%)	
Total FSH dose (IU)	3 045.5 ± 1 065.3	2 006.3 ± 608.6	<0.001

Body mass index (BMI), anti-Mullerian hormone (AMH), antral follicle count (AFC), follicle-stimulating hormone (FSH), luteinizing hormone (LH), estradiol (E2), gonadotropin releasing hormone (GnRH).

**Table 2 jcm-11-07173-t002:** IVF cycle outcomes for patients with Diminished Ovarian Reserve (DOR) compared to patients with Normal Ovarian Reserve (NOR). Values are shown as mean ± SD or *n* (%).

	DOR (*n* = 76)	NOR (*n* = 180)	*p*-Value
Treatment type			0.006
IVF	48 (63.2%)	80 (44.4%)	
ICSI	28 (36.8%)	100 (55.6%)	
Semen parameters			0.03
TMSC ≥ 5 millions	55 (72.4%)	105 (58.3%)	
TMSC < 5 millions	21 (27.6%)	75 (41.7%)	
No. of oocytes	6.0 ± 3.2	12.7 ± 6.3	<0.001
No. of embryos	3.2 ± 1.9	6.0 ± 3.7	<0.001
Embryo transfer			0.03
Single embryo transfer	20 (28.6%)	74 (43.3%)	
Double embryo transfer	50 (71.4%)	97 (56.7%)	
Stage at transfer			
Day 2	29 (41.4%)	63 (36.8%)	0.06
Day 3	37 (52.9%)	78 (45.6%)	
Blastocyst	4 (5.7%)	30 (17.5%)	
Biological pregnancy	20 (28.6%)	55 (32.2%)	0.6

In vitro fertilization (IVF), intra cytoplasmic sperm injection (ICSI), total motile sperm count (TMSC).

**Table 3 jcm-11-07173-t003:** Absolute and relative morphokinetic events of embryos from patients with Diminished Ovarian Reserve (DOR) compared to patients with Normal Ovarian Reserve (NOR). Values are shown as mean ± SD.

Morphokinetic Parameter (h)	DOR (*n* = 242)	NOR (*n* = 1072)	*p*-Value
t2	5.7 ± 9.6	5.5 ± 9.8	0.8
t3	14.5 ± 8.8	14.0 ± 9.7	0.5
t4	16.5 ± 8.4	16.3 ± 9.5	0.7
t5	24.9 ± 9.8	25.3 ± 10.5	0.6
t8	33.3 ± 6.7	32.4 ± 7.6	0.3
tB	89.2 ± 13.9	87.7 ± 10.7	0.4
ECC2	11.7 ± 5.9	11.3 ± 5.2	0.3
cc2a	9.3 ± 5.8	9.0 ± 5.4	0.4
ECC3	18.5 ± 5.3	17.7 ± 5.8	0.2
s2	2.4 ± 4.9	2.4 ± 4.9	>0.9
s3	9.1 ± 6.8	8.5 ± 6.9	0.5

Time to two-cell (t2), three-cell (t3), four-cell (t4), five-cell (t5), eight-cell (t8), time to full blastocyst (tB), duration of the second cell cycle (ECC2 = t4 − t2, cc2a = t3 − t2), duration of the third cell cycle (ECC3 = t8 − t4), the synchronicity of the divisions within the second cell cycle (s2 = t4 − t3), the synchronicity of the divisions within the third cell cycle (s3 = t8 − t5).

**Table 4 jcm-11-07173-t004:** Absolute and relative morphokinetic events of embryos in the two different age groups. Values are shown as mean ± SD.

Morphokinetic Parameter (h)	Age < 37 (*n* = 853)	Age ≥ 37 (*n* = 461)	*p*-Value
t2	5.4 ± 9.7	5.9 ± 9.7	0.3
**t3**	**13.7 ± 9.2**	**14.9 ± 10.0**	**0.02**
**t4**	**15.8 ± 8.7**	**17.3 ± 10.3**	**0.006**
**t5**	**24.8 ± 10.1**	**26.2 ± 11.0**	**0.04**
t8	32.5 ± 7.6	33.1 ± 7.2	0.3
**tB**	**86.9 ± 10.4**	**90.1 ± 12.4**	**0.008**
**ECC2**	**11.1 ± 5.0**	**11.8 ± 5.9**	**0.02**
cc2a	8.9 ± 5.4	9.2 ± 5.6	0.4
ECC3	17.9 ± 5.8	17.9 ± 5.5	0.9
s2	2.4 ± 5.1	2.6 ± 4.6	0.5
s3	8.6 ± 6.9	8.7 ± 6.9	>0.9

Time to two-cell (t2), three-cell (t3), four-cell (t4), five-cell (t5), eight-cell (t8), time to full blastocyst (tB), duration of the second cell cycle (ECC2 = t4 − t2, cc2a = t3 − t2), duration of the third cell cycle (ECC3 = t8 − t4), the synchronicity of the divisions within the second cell cycle (s2 = t4 − t3), the synchronicity of the divisions within the third cell cycle (s3 = t8 − t5). Significant *p*-Values (< 0.05) are shown in bold.

**Table 5 jcm-11-07173-t005:** Results of the generalized linear mixed-effects models regarding the effect of age on embryo morphokinetic parameters.

Morphokinetic Parameter (h)	Beta Coefficient (SE)	*p*-Value
t2	0.002 (0.07)	>0.9
t3	0.07 (0.1)	0.5
t4	0.6 (0.8)	0.4
t5	0.2 (0.9)	0.8
t8	0.2 (0.7)	0.8
tB	2.1 (1.8)	0.2
ECC2	0.1 (0.07)	0.1
cc2a	0.03 (0.1)	0.8
ECC3	−0.1 (0.6)	0.8
s2	0.06 (0.04)	0.1
s3	0.06 (0.7)	>0.9

Time to two-cell (t2), three-cell (t3), four-cell (t4), five-cell (t5), eight-cell (t8), time to full blastocyst (tB), duration of the second cell cycle (ECC2 = t4 − t2, cc2a = t3 − t2), duration of the third cell cycle (ECC3 = t8 − t4), the synchronicity of the divisions within the second cell cycle (s2 = t4 − t3), the synchronicity of the divisions within the third cell cycle (s3 = t8 − t5). Beta coefficients (β), Standard Estimates (SE), and *p*-values are shown for each parameter.

## Data Availability

The data presented in this study are available on request from the corresponding author.

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
