# Peer review of "Influence of Diminished Ovarian Reserve on Early Embryo Morphokinetics during In Vitro Fertilization: A Time-Lapse Study"

_jcm, 2022, doi:10.3390/jcm11237173_

Round 1

Reviewer 1 Report

The authors have analysed the impact of Diminished Ovarian Reserve on morphokinetic patterns of embryo development. In general, the manuscript is well-written and contains some interesting ideas and new information.

Author Response

Not applicable.

Reviewer 2 Report

The study design is well done, the descriptions are accurate, as are the literature references.

The results are clear and accurate. One suggestion might be to also include information regarding men such as age, whether they smoke etc. in the section where outcomes for NOR and DOR patients are compared.

All in all, it turns out to be a good paper with good scientific impact.

Author Response

We thank the reviewer for this suggestion. Paternal age and smoking status have been added Table 1. Since morphokinetic parameters were not significantly different between the DOR and NOR groups, there was no need to include paternal age and smoking status in a multivariate analysis.

Reviewer 3 Report

- in each table  - the results should be presented as mean values  ± SD (instead of SD in parentheses),  it will be more clear  for the readers

- the results presented in the tables should not be repeated in the text: pages 5 - you could remove all mean values ± SD (given in parentheses) from the text described above the tables  1 and 2

 - It is not clear how age groups (< 37 and ≤ 37) correspond to DOR and NOR groups, it should be clarify, included in the table 4 and also discussed in this context

 - age of the male partner should be given for both  groups

 - line 293 - Schachter-Safrai  - the number of reference is missing
